# Why a mechanistic theory of soils is crucially important: Another line of supportive argument exists, seldom invoked in soil science

Philippe C. Baveye[1]

[1] Saint Loup Research Institute, 7 rue des chênes, La Grande Romelière, 79600 Saint Loup Lamairé, France.

*Correspondence to*: Philippe C. Baveye (Philippe.Baveye@Saint-Loup-Institute.org)

**Abstract.** In the last few decades, the effort that has been devoted to the mechanistic, quantitative description of soil processes has been justified on the grounds that theories and models help us understand how soils function, and also predict how, e.g., they are likely to adjust in the future to environmental change. The argument, familiar to physicists, that theories uniquely determine what should be measured, has rarely if ever been invoked in the soil science literature. On the contrary, to enable the classification and mapping of soils, enormous amounts of "theory-free" data have been and continue to be amassed by soil scientists. In this general context, the key objective of the present Forum article is to argue that the accumulation of more "theory-free" data, in particular to allow the application of artificial intelligence methods, does not make practical sense at this stage, and that the development of improved theories of soil processes is crucial to provide guidance about the type of measurements that should be collected. Hopefully, this Forum article will stimulate a debate on this issue, and will lead to a much-needed intensification of theoretical research and modelling in soil science.

## 1 Introduction

Over the last few decades, research efforts have been devoted to the development of mechanistic theories describing some of the physical, (bio)chemical, and biological processes taking place in soils (see, e.g., the review by Vereecken et al., 2016). These theories have led to the emergence of a large array of computer models, using a variety of numerical approaches. Several authors, in recent years, have called for an intensification of the research in this area, toward the development of a "theory of soils" (e.g., Neal, 2021; Neal et al., 2021).

In their justification for the need for theory and model development, researchers in soil science have traditionally relied on one of two lines of argument. The first is that by trying to encapsulate into a mechanistic theory what we know about a given process and by comparing computer model outputs with experimental data, it is possible to assess whether our knowledge appears to be satisfactory or is missing some important component of reality. From this perspective, theory building assists the discovery process. Another argument is that theories, and especially the computer models they lead to, help us predict how soil processes are likely to evolve in the future, in the context of climate change (e.g., Chalhoub et al., 2025) or when we try to determine which soil management practices are beneficial to address mounting demands on soils (e.g., Ray et al., 1996).

An additional line of argument, commonly adopted in other disciplines, appears to have rarely made any noticeable foray into the soil science literature. It considers that mechanistic theories are absolutely necessary to determine what needs to be measured. This perspective is frequently embraced in physics, among other fields of knowledge. Decades ago, Einstein (quoted in Salam, 1990) opined that theory "decides what can be observed", and by extension, what cannot. For instance, Heisenberg's indeterminacy principle states that one cannot measure both the position and momentum of a particle, such as a photon or electron, with perfect accuracy; one has to choose one or the other. In addition to spelling out clearly what can and cannot be measured, a theory also points out the parameters it makes sense to measure, and, by default, those that are irrelevant. For example, because the theory we rely on to accurately describe the thermal expansion of metals under usual environmental conditions involves ambient temperature but not the $CO_2$ concentration of the air, we know that measurements of the former are crucial to predict in practice by how much above-ground electrical cables are likely to elongate in the summer, whereas measurements of the latter are not useful at all.

In this general context, the primary objective of this Forum article is to stimulate a debate on the perspective that mechanistic theories are crucially important in soil science as well, to determine what characteristics of soils we need to measure to manage soils properly, and to design appropriate experiments to test specific hypotheses. To put the narrative in a proper framing, I first try to understand why, in soil science, this line of argument in support of theory development appears to have so rarely been alluded to.

## 2 Historical reliance on "theory-free" data in much of soil science

When, in mid-19[th] century, scientists started paying serious attention to soils, they were immediately confronted with their enormous spatial heterogeneity, requiring special ways to classify and map them. To that end, given the sizeable task at hand, measurement methods were needed that at the same time were rapid, could be implemented directly in soil pits or in a rudimentary field laboratory without requiring sophisticated equipment, and focused on soil properties that, one could reasonably assume, did not change appreciably over time. The particle size distribution of soils, the depth, color, and organic matter content of their horizons, or the average size of "aggregates" identified after destructively sampling soils, were among these easily measured properties and have constituted the basis on which soils have been mapped for over a century and a half. Their measurements do not require any underlying mechanistic theory of soil processes, either to design the data-acquisition methodology or to interpret what the data imply. Until the 1960s, when computers started being used extensively to try to describe and predict soil behavior, there was very little theory of soil processes to speak of anyway, which could have been used to design measurement methods, except in relation to the movement and retention of water in soils. However, the associated measurements were far too laborious and time-consuming to be used for soil mapping over large regions.

When, after the 1960s, soil survey campaigns got completed in a number of countries (or were simply terminated in others, like France, for lack of support), and interest among soil scientists tended to shift to the multitude of soil processes

responsible for the multifunctionality of soils (Simonson, 1966; Heuting, 1970), it became rapidly evident that "theory-free" soil data gathered for the purpose of soil classification and mapping would no longer be adequate. A vivid example of that involves the cation exchange capacity (CEC) of soils, which, for purposes of soil classification, had traditionally been measured via replacement of exchangeable cations with a saturating solution at a set pH of 7 or 8.2 (e.g., Sumner and Miller, 1996). This standard measurement technique produces results that do not make much sense when one attempts to describe

soil processes, because under undisturbed conditions in soils, the pH often differs from those set values, and because, as shown experimentally by a number of researchers (see, e.g., Mokady and Bresler, 1968; Boast, 1973; Barton and Karathanasis, 1997), the actual CEC of a partly water-saturated soil differs from that obtained when the soil is fully water-saturated.

Nevertheless, the idea emerged in part of the soil science community that the mass of "theory-free" data that by then had

75 been accumulated could still be useful to describe soil processes, provided one could establish sufficiently strong statistical correlations between these data and the various "difficult to measure" dynamical parameters needed in that context. In spite of the many questions this idea raises, since it is well known in statistics that correlation and causation are two very different things, and that the former does not imply the latter, it seemed reasonable for some researchers to try to develop a number of such statistical correlations, termed "pedotransfer functions" (see, e.g., Bouma, 1989; Bouma et al., 1996, 2022; Lin, 2003;

Pachepsky et al., 2006; Weijnants et al., 2009; Vereecken et al., 2010). Once this effort got underway, it served to legitimate the continued acquisition of "theory-free" data as well as the recent strong push to develop large repositories of soil data, like the Land Use and Coverage Area frame Survey (LUCAS) database (Orgiazzi et al., 2018), as well as to digitize and interpolate soil maps globally to make them more accessible to users. It also fueled the motivation of some soil researchers to advocate that research on soils could be "data-driven", e.g., using data-mining, machine-learning, or Artificial Intelligence

(AI) techniques (Bui et al., 2009; Bui, 2016; Chen et al., 2019; Wadoux et al., 2021; Wadoux, 2025; Minasny and McBratney, 2025; Teodosio et al., 2025; Hashimoto et al., 2025).

These efforts were, and still are, largely inspired by the need for parameters for land surface or hydrological models for which we require basin-wide, regional, or even global parameter maps. Recently "covariate-based Geo Transfer Functions" (co-GeoTF) (e.g., Gupta et al., 2022) have been introduced to replace pedotransfer functions, so that not only soil- but other

environmental variables (climate, topography) as well are considered, in order to explain the variability of soil parameters. Also, some researchers have advocated for the use of "physics-informed" machine learning (PIML), and in particular of "physics-informed neural networks" (Wang et al., 2023; Norouzi et al., 2025), which perhaps signals a trend in the estimation of soil parameters to move away from pure data-driven statistical approaches toward attempting to combine some theory (in this case about soil water retention) with machine learning methods.

There is a risk that the above remarks might be interpreted as a kind of scolding of those who engage(d) in the development of pedotransfer functions or co-GeoTFs. That is not my intention at all. It would be like blaming frogs in the well-known boiling frog apologue. Once the movement got underway to try to make as much use as possible of available soil data, regardless of the fact that they had been originally obtained for a very particular purpose and may not be directly relevant to

other goals, soil researchers undoubtedly must have found it difficult to react against the rising tide and jump out of the boiling water, metaphorically, especially given that, at the time, there was little concomitant effort underway to develop an alternative, theory-grounded perspective. The more emphasis was placed on exploiting freely available "theory-free data" and on gathering additional ones, the more it became unavoidable for researchers to feel compelled to use pedotransfer or GeoTransfer functions to obtain the basin-wide, regional, or even global parameter maps required to populate the input parameters needed for land surface or hydrological models, for example. One might argue in this respect that if hydro-thermal parameter maps (e.g. Montzka et al., 2017; Dai et al., 2019) had not been developed, land surface modellers might still be using look-up tables of hydraulic and thermal parameters, encompassing only a handful of soil types (Weihermueller et al., 2021).

Fundamentally, one could argue that, over the years, the focus of many researchers in soil science on "theory-free" data and on potentially groundless statistical correlations, rather than on trying to understand and describe quantitatively the basic mechanisms involved in soil processes, allowed the proliferation of what one of the reviewers of the present article refers to as "myths". Examples abound and include, to name only a few, the alleged positive impacts of biochars on soil structure and crop yields (e.g., Wang et al., 2017; Jeffery et al., 2017), of soil biodiversity and a number of other "indicators" on soil functions/services (e.g., Pulleman et al., 2012), or of some aspects of the metagenomic make-up of soils on the activity of microorganisms (e.g., Myrold et al., 2014). For example, if researchers had from the onset tried to understand at a basic level by what mechanisms biochars might improve crop yields, they might very well have discovered that in many situations biochars by themselves do absolutely nothing, but that it is the various nutrients that biochars contain initially as a result of their pyrolysis that have an effect on crop yields, and only for a short period of time. In order to ascertain this, a full characterization of the physical and chemical properties of biochars (e.g., Baveye, 2014; Schnee et al., 2016), clear testable hypotheses about mechanisms, and targeted experiments with proper controls would have been required. A recent analysis of the literature on the topic shows that, largely, this research has yet to be carried out and, because of that, no real conclusion can be drawn at present (Chaplot et al., 2025). The same applies to the extent to which the incorporation in soils of fresh organic matter, e.g., cover crops, really contributes to the long-term "sequestration" of carbon in soils (e.g., Chaplot and Smith, 2023, 2024; Baveye et al., 2023). Another example concerns the concept of "aggregates" in soils, which some authors consider absolutely essential to describe the functioning of soils, even though no mechanistic theory or model links aggregate characteristics with any process taking place in soils, and aggregates cannot be readily identified in undisturbed soils (e.g., Garland et al., 2024; Vogel et al., 202; Baveye et al., 2022, 2024). More myths like those have been discussed in a number of other articles published in the last few years (e.g., Baveye and Wander, 2019; Baveye, 2021a,b,2022; Baveye et al., 2024), At another level, one might also argue that the fact that many researchers are content with merely identifying correlations between parameters explains why "soil health" remains so poorly conceptualized at this point (e.g., Baveye, 2021; Harris et al., 2022; Baveye et al., 2025): Any attempt to come up with a theory or model of soil health would require as a prerequisite that the concept be defined mechanistically, which is still not the case at this stage.

Some researchers harbour great hope that machine learning and AI will help address and answer successfully some of the key questions about soils with which we are confronted. However, recent research points out various drawbacks of these approaches and raises questions about the fact that they are widely viewed as a panacea at the moment (e.g., Wadoux, 2025;
Hashimoto et al., 2025). The very recent article of Uxa (2025) deals with a specific situation where "statistical learning violates physics". He shows that the use of global soil temperature maps, generated using machine-learning at a 1-km$^2$ resolution for two depth levels of 0–5cm and 5–15cm, can lead to unacceptable outcomes, namely "reversed patterns between the two depth levels in terms of soil temperature physics, with more pronounced temperature amplitudes, minima, and maxima at the deeper level, which has no reasonable physical explanation." Because a theory exists in this case, dealing
with the propagation of diurnal temperature fluctuations in the subsurface, it is possible to determine that the maps machine learning and AI are producing are not sound. In the absence of an underlying theory, it is not clear that errors would have been identified as easily.  Physics-informed machine learning (e.g., Wang et al., 2023; Norouzi, 2025), taking advantage of the fact that a theory exists to describe the retention of water by soils, may help alleviate some of these pitfalls. However, the problem persists for other dynamical aspects of soils for which no fundamental theory is as yet available.

Another very serious issue with machine learning and AI directly relates to their reliance on correlations. Because of it, it is not clear at this stage whether these approaches still hold the same "potential for advancing knowledge and innovation" (Wadoux, 2025) they were claimed to have not very long ago. Indeed, they have clearly run into a serious snag.  Starting with Fourcade et al. (2018), various authors have demonstrated that statistically stronger patterns can emerge from databases to which one has deliberately added entirely irrelevant information, for example, a painting or the photograph of a colleague
(Behrens and Viscarra Rossel, 2020; Wadoux et al., 2020; Rentschler and Scholten, 2025). Clearly, machine-learning and AI techniques are not able at all in this case either to discriminate on their own between what is meaningful information and what is not, and therefore do not appear likely to "uncover the missing pieces of process-based models", as recently claimed (Hashimoto et al., 2025), except perhaps fortuitously. This is hardly surprising since these methods rely heavily on correlations, not causation. Rentschler and Scholten (2025) recently concluded from it that users have the responsibility to
make sure that parameters they consider when implementing these techniques are "in line with existing scientific theory of *mechanistic and process understanding*" (emphasis added). In other words, after decades of trying with a variety of statistical and, lately, AI approaches to rely on masses of "theory-free" data to describe soil processes and functions instead of spending time developing dedicated theories or computer models, it appears necessary to backpedal, and to address head-on the fundamental question of what data we actually need, i.e., which ones are directly relevant to what we want to do.

**3 A theory is needed to determine which measurements are relevant**

Perhaps the most persuasive plea for the crucial importance of theory in the context of the description of soil processes comes from the literary world. In her 1923 novel "Murder on the Links", which I think should be recommended reading for all soil science students, Agatha Christie contrasts the modi operandi of two detectives. A French detective runs around

feverishly, painstakingly amassing all kinds of information, whereas the famous Belgian detective Hercule Poirot (who, of course, ends up solving the case) mostly sits in an armchair, trying to elaborate a theory of the murder. Poirot's philosophy is that, past a certain point in a murder investigation, one does not know whether a given bit of information is a clue or, even more basically, where to look to gather additional information in a time-efficient way, unless one has a guiding theory.

The early history of soil physics provides a vivid example of the soundness of this perspective. Buckingham (1907), starting from first principles in physics, developed a theory of water movement in unsaturated soils in which traditional easy-to-measure, "theory-free" physical properties, like soil texture or aggregate size, were notably absent. Instead, his theory identified for the first time the soil water matric potential, the soil water retention curve (SWRC), and the unsaturated hydraulic conductivity as essential to the description of water movement (e.g., Nimmo and Landa, 2005; Narasimhan, 2007). At the time, equipment was entirely lacking for their measurement in the field. The first tensiometer, enabling the measurement of the soil water matric potential, was developed a year later, in 1908 (e.g., Or, 2001). The next half century saw the development of a whole panoply of instruments, for use both in the laboratory and under field conditions to characterize the physical status of soils.

I will argue in the following that this early history of soil physics could serve as a blueprint for how to proceed in other areas of soil science. To be honest, however, and to avoid giving the impression that I put soil physicists on a pedestal, one has to acknowledge that the subsequent evolution of that discipline has not always adhered to the same epistemological standards. Over time, and as already discussed above, some soil physicists have also fallen prey to the temptation of connecting the reputedly "hard to measure" physical parameters to easily measured ones like soil texture via pedotransfer or other empirical functions, in spite of Buckingham's (1908) clear theoretical demonstration that texture is not pertinent to the description of water transport and retention in soils. Moreover, more than 30 years after Letey (1991) advocated convincingly that the concept of soil aggregate was not useful to describe the structure or architecture of soils, quite a few soil physicists still continue to maintain that consideration of soil aggregates is essential to describe soil processes, as illustrated by the exchanges that have followed the publications of Vogel et al. (2022) and Baveye et al. (2022). One could also argue that the fixation of soil physicists on geostatistics and fractal geometry in the 1980s and 1990s, respectively, was related more to the appeal and aesthetic of these tools than to a careful consideration of what the key questions were that they could help address (see discussion in Baveye and Laba, 2015; Baveye et al., 2024).

Be that as it may, it is instructive to find out how the Buckingham blueprint could be applied in other areas of soil science, where sound theories are still lacking. For instance, there is a lot of interest currently in the effect that bacteriophages could have on various soil processes, like the mineralization of soil organic matter (SOM) and the resulting emission of greenhouse gases by soils (e.g., Pratama and Van Elsas, 2018; Lee et al., 2022; Hazard et al., 2025). Estimates are that there are 10 times as many bacteriophages as bacterial cells in soils, so their —so far largely unknown— overall effect could be significant. Should we account for it explicitly when we try to predict the fate of SOM? To answer this question, the traditional approach in soil science would have us measure everything we can about bacteriophage abundance and diversity in soils, after which we would try to determine statistically if these data are correlated significantly to processes of interest. This would require an

enormous amount of work, likely spanning many years and forcing us to spend a lot of time and money measuring a large number of parameters that might turn out eventually not to matter. By contrast, a theoretical approach would consist of developing a model of bacteriophage action in soils, based, e.g., on what is known about the behavior of viruses in other systems, and about the dispersion in soils of nanoparticles similar in size to phages. The hypotheses embodied in this model could be tested in simple targeted experiments involving one or two bacteria-bacteriophage pairs, either in actual, sterilized soil samples, or in 2D micromodels. Once tested, the model could then be incorporated with the description of other aspects of soils (see Figure 1) to obtain a comprehensive theory. In many ways, the same approach should be adopted with respect to taking soil fauna explicitly into account in the description of soil processes (Briones, 2014, 2018; Cayuela et al., 2020).

Realistically, the development of this comprehensive theory is not going to be straightforward. Many questions still remain unanswered about the proper framework for this work. Ecosystem-scale models have assumed for a long time that SOM could be apportioned into separate pools with distinct turnover times, and that its mineralization kinetics could be described by simple first-order reactions, without explicitly accounting for the presence or activity of microorganisms. Efforts have been made in the last two decades to introduce features in the models that deal explicitly with microbial action, carbon use efficiency, and priming, but proponents of this approach admit that they are still largely struggling with it (e.g., Schimel, 2023). A very different, "bottom-up" approach, stimulated by the commercialization of table-top X-ray scanners twenty years ago as well as progress with a number of spectroscopic and pore-scale modelling techniques, consists, as in Figure 1, of developing a model of soil carbon dynamics on the basis of observations of soil processes at the microscale, commensurate with the scale at which microorganisms operate (e.g., Baveye et al., 2018; Pot et al., 2021, 2022a,b). A serious challenge in this context is the need to upscale the description of processes from the micro- to the macroscale, although this still largely unresolved upscaling hurdle occurs no matter which approach, top-down or bottom-up, is adopted (Baveye, 2023). Another challenge is that some of the parameters that the microscale modelling indicates should be measured, like the average spatial separation between microorganisms and SOM (e.g., Mbé et al., 2022), cannot be measured directly at the moment, so that alternative routes to these parameters need to be found.

One of the reviewers of this article, referring to past examples in physics and astronomy, suggested that a good way to proceed toward the development of a mechanistic theory of soils would be to first list the various "myths" currently being circulated in our discipline, to get an idea of what the theory should look like and what it should be able to explain. Right or wrong, I tend to believe that listing all, or even a portion, of the numerous "myths" currently circulating about soils would be a dauntingly time-consuming endeavour. Nevertheless, focusing on some of them might indeed be a good way forward. In essence, one might consider that this is precisely what motivated the work on the microscale description of soil processes over the last two decades (e.g., review in Baveye et al., 2018). Realization that, contrary to prior belief, traditional kinetic equations involving macroscopic measurements of SOM content and of soil biomass could not help us predict how much actual mineralization of SOM would take place in the few years following the addition of organic residues to soils, prompted a number of us to try to quantify the microscale spatial distributions of SOM and microorganisms, respectively. A particular difficulty with this type of research is caused by the fact that, in soils at the microscale, physical, mineralogical,

(bio)chemical, and biological processes interact intimately, and cannot be meaningfully studied separately. One could arguably reach the same conclusion about any of the other myths in soil science.

To make progress, research therefore needs to be interdisciplinary, which implies a break with the very much mono-disciplinary focus that continues to dominate soil science (e.g., Baveye and Wander, 2019; Baveye et al., 2024). We need to train soil scientists who are equipped to engage in this type of research. One could argue that this is still not the case, even though the topic has been discussed for decades (e.g., Wagenet et al., 1992). In addition, the development of a comprehensive theory of soils will require soil scientists to be far more quantitatively literate than they tend to be, except for mathematically-minded soil physicists. Perhaps one of the reasons there has been so much emphasis on statistical correlations in soil science up to now is that, to a large extent, they correspond to the level of mathematical complexity that most soil scientists are comfortable with. That of course does not mean that statistical correlations are somehow inferior to the equations of mathematical physics, but it is simply an observation that the mathematics involved in correlation formulas is clearly far easier to deal with than ordinary or partial differential equations, or some of the advanced geometrical representations of the pore space currently used in microscale computer models of soil processes (e.g., Kemgue et al., 2019; Pot et al., 2021, 2022a,b). As pointed out by the other reviewer of this article, Professor Göran Ågren, and as illustrated clearly in Bosatta and Ågren (1991, 2003) and Ågren (2021), "to get from the many conventional discrete pool models of soil organic matter transformations to models based on process understanding raises the level of mathematical knowledge considerably".

As Professor Göran Ågren suggests, one possible solution to this problem might be to involve soil physicists closely in any future research effort on soils requiring some form of quantitative description. However, I would contend, based on my experience over the last four decades, that interdisciplinary research yields valuable results only if all participants share a common language. It would not do in this context if at the point of describing things quantitatively, participants just turned to soil physicists and told them "OK, now you do your magic, and it's a wrap…" In that respect, there is a clear need to rethink the training of the next generation of soil scientists, to make it more genuinely interdisciplinary from the start, for example by using a "problem-based" approach, and by not neglecting quantitative, theoretical approaches. Some of us have been consistently advocating for that over the last 30 years (e.g., Wagenet et al., 1992; Amador and Görres, 2004; Baveye et al., 2006, 2014, 2024).

## 4 Take-home message

In this forum piece, I have argued that soil scientists should discontinue the long-standing practice of accumulating masses of "theory-free" data and of attempting to merely correlate them to processes of interest. Numerous examples in physics, including in soil physics with the work of Buckingham (1907), show that we need theories to determine which parameters have to be measured to describe dynamical soil processes quantitatively, and ultimately manage them properly. We have some theories already for specific situations, and significant progress has been achieved in recent years toward the

development of a comprehensive theory of physical, (bio)chemical, and microbiological processes in soils, but a lot more work remains to be done in the area, in particular to better take into account a number of aspects that have been neglected in that context so far, like plant-soil relationships, and the effect of soil fauna or bacteriophages. The hope at this stage is that, like in ecology a decade ago following a call by some researchers for more theoretical grounding of the discipline (e.g., Niemelä, 1999; Scheiner and Willig, 2008; Marquet et al., 2014), the sustained debate this article is trying to stimulate on the need to develop new theories in soil science will help determine which theoretical framework is most suitable for this work, and will convince soil scientists that a significant interdisciplinary effort is in order rapidly. At a time where reliance on AI techniques is encouraging researchers to gather increasingly massive amounts of soil data, and as large data repositories, like the LUCAS database, are being set up, any progress made on the theoretical front should help guide sampling efforts and assess what measurements would actually be needed.

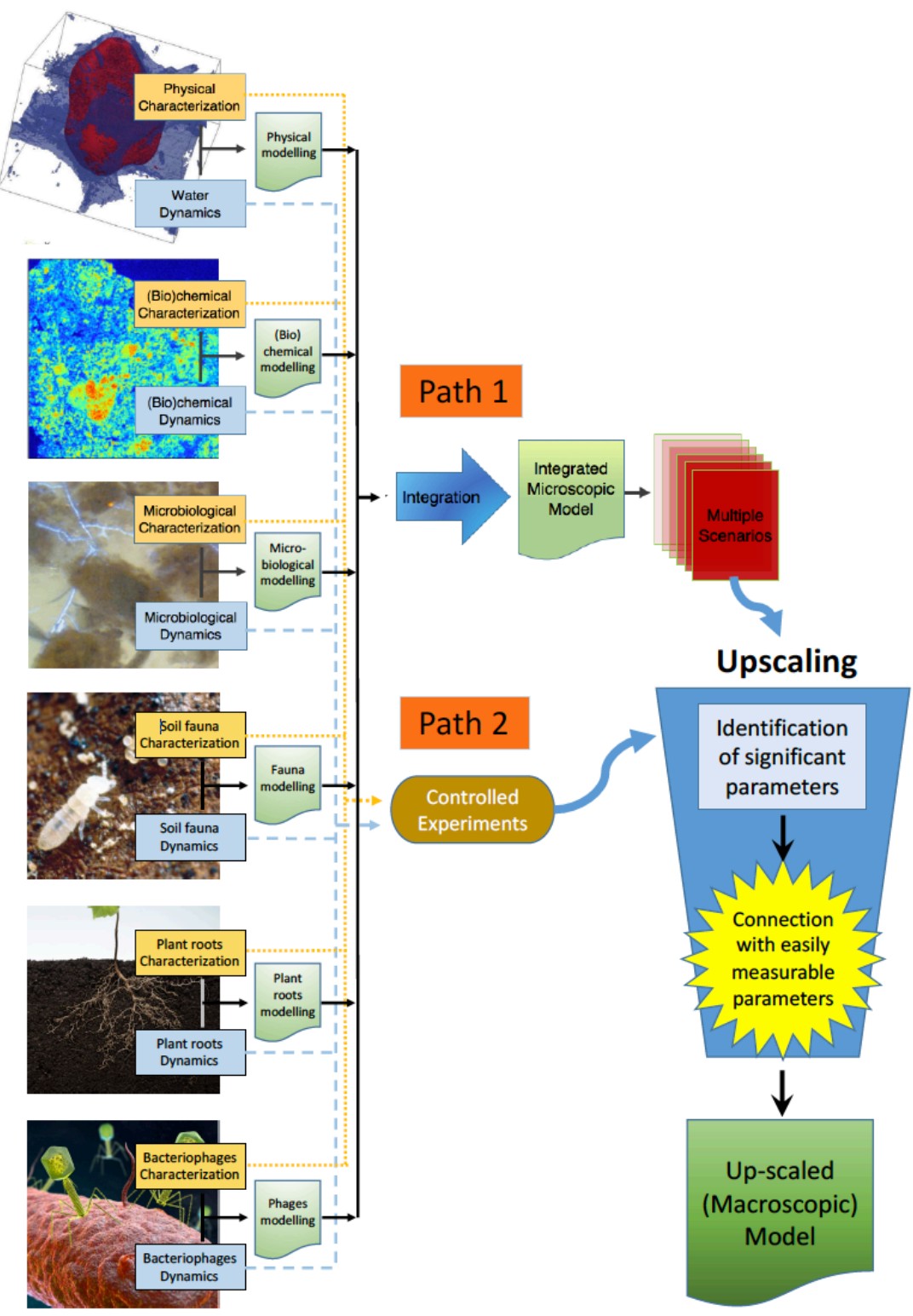

**Figure 1: Schematic representation of a possible sequence of steps in the development of a model of soil carbon dynamics applicable to large (macroscopic) spatial scales, starting on the left from a characterization of the static (light brown boxes) and dynamic (light blue boxes) components of different properties of soils at the microscale. The green boxes correspond to initial (left), intermediate (center), and final (right) modelling efforts. Following path 1, the integration of different perspectives results in the development of a microscale model of soils, which can be run multiple times, under a variety of scenarios. Along path 2, controlled experiments are carried out to obtain macroscopic data against which microscale measurements on soil samples can be contrasted. These data, in parallel with the outcomes of scenario modelling, feed into the upscaling step, whose goal is to identify easily measurable macroscopic parameters associated with an up-scaled model (Baveye, 2023).**

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
