# Peer review of "Why a mechanistic theory of soils is crucially important: Another line of supportive argument exists, seldom invoked in soil science"

_EGUsphere, 2025_

## Referee Comment (RC1)

**Baveye: Why a mechanistic theory of soils is crucially important: Another line of supportive arguments exists, seldom invoked in soil science**

**General comments**

What the author argues about is rather obvious. Having a theoretical basis for performing experimental studies makes them more structured and efficient. This paper can hopefully inspire some scientists to take some time to develop theories before embarking on extensive experimental studies. A problem I see here is the weak mathematical training soil scientists have. As an example, to get from the many conventional discrete pool models of soil organic matter transformations to models based on process understanding raises the level of mathematical knowledge considerably ( see papers on a continuous description rather than discrete ones, e.g. Bosatta, E. Ågren, G I. 1991. Dynamics of carbon and nitrogen in the organic matter of the soil: A generic theory. *The American Naturalist* 138: 227-245). A possible solution to this problem could be to invite physicists to participate in soil science studies.

---

## Author Response (AR1)

**Itemized response to Editor's comments**

This Forum article has an important message to tell, and I fully agree with its observations about how to train the next generation of soil scientists in relation to mathematical prowess for mechanistic theory development, and a much stronger emphasis on interdisciplinarity. However, the author needs to make sure that the arguments remain balanced and do not sound like a scolding of those of us who engage in the development of pedo-transfer functions (in the widest sense of the word) and machine learning techniques, although I fully take on board the improvements proposed in those areas (but note that most of those are already ongoing, see below). To populate the parameters of land surface or hydrological models we require basin-wide, regional or even global parameter maps. So, until we have suitable alternatives we will have to rely on pedo-transfer functions, or preferably co-variate geotransfer functions (see e.g. Gupta et al., 2022).

> **Author's response**: I have tried as well as I can to address the "scolding" comment on lines 83 to 92 of the re-revised text. I do not want anyone to have the impression that they are being "scolded". What I have tried to convey is what I felt in the 80s when every article, every proposal, in soil physics had to be about geostatistics to be relevant, it seemed. And again in the 90s, same story about fractal geometry. It is like being in a tide, from which one cannot escape… I still feel, personally, that the development of pedo- and geotransfer functions was a mistake, and I am glad I have never had to actually use one in my research, but once the movement was underway, I am fully aware of how difficult if not impossible it must have been for most people to go against it.
> In the previous paragraph (lines 75 to 77), I have also made reference to the two articles by Gupta et al. and Norousi et al. that you mentioned.

In the lines below I am referring to the revised article you shared with me. If you could bring in and address the various issues I mention below, then I am happy to accept the article, because you have already dealt nicely with the comments of the reviewers.
-Line 64/65: It may be worth mentioning the nearly finalised EU Soil Monitoring Law in places, which will involve new soil sampling efforts, and more recent sampling effort related to the LUCAS data base, for example. New soil monitoring is still going on? Indeed, it needs informed decisions to guide these sampling efforts, and your article can be helpful in that respect

> **Author's response**: Indeed! I now mention the Lucas database explicitly on lines 78-79 and 278-280.

-Line 94: It is the physical aspects of biochar that also play a role, via the properties of this porous medium (e.g. its specific surface areas and pore size distribution), that affect water retention, not just the chemical properties. There are plenty of papers on this that may deserve a mention?

> **Author's response**: I agree. On line 108, I added a citation to an article by Schnee et al. that analyses in detail the physical side of things. I refrained from adding many more references, because I do not want the focus to be too much on biochars…

- Around line 120 and see also line 130-137: it would be good to mention physics informed machine learning, e.g. physics informed neural networks (PINNS), see e.g. Norouzi et al. (2025) and references herein. What you are discussing here is mostly data-driven ML. PIML would do a much better job at this and probably weed out those 'images of a colleague'.

> **Author's response**: I am not sure I share your enthusiasm for PINNS… From a theoretical point of view, it is well known that any attempt to relate soil texture with the SWRC is necessarily doomed in general. I have worked on soils in Costa Rica that were 95% clay and behaved like coarse sands. Conversely, in my research on the bioclogging of sands, bacteria could make coarse sands have the saturated hydraulic conductivity of fine silts… Without going into details, I mention on lines 162-165 the lack of relevance of soil texture to the description of soil water transport and retention. Not everyone will agree with me on that, and that is OK. Perhaps my Forum piece will encourage some to engage in a constructive debate on this and other topics I cover.

- Line 197-199: The text on "myths" is perhaps somewhat overly negative? Are there really that many myths in soil science ("dauntingly time-consuming endeavour")? Can you give some examples other than biochar? Aren't we suffering from the fact that soil is so closely related to agronomic practices, and when trying to feed the world farmers are quick to adopt practices that improve their yield, and then many pseudo soil scientists get in on the action which then leads to 'proper' soil science getting a bad name. The use of liquid nano-clay, to improve desert soils, is another example.

> **Author's response**: I wish I could believe, like you do, that there aren't that many myths in soil science. But I feel that there are way too many. I added a short mention of another, related to aggregates, on lines 113-116. And then I pointed out a number of other articles I have written over the last 10 years in which readers could find several other examples.

- Again, the tone could be softened a little around lines 214-230 of the revised article you shared with me: e.g., "the level of mathematical complexity that most soil scientists can handle". Also, this paragraph makes it sound like knowledge of detailed statistical theories is inferior to physical/mathematical science. I think there is room for both, and they should go hand in hand.

> **Author's response**: I agree that my original text may have given the impression that you got upon reading… I have modified the text on lines 247-250 to reflect that. And I agree that there should be room for both, as you write. It is unfortunate that, in the training of soil scientists, there is generally little room spared for the slightly more elaborate mathematical descriptions of soil processes. I remember giving a talk once to an audience of soil microbiologists, to which I tried to explain that there were good reasons to think that the Monod equation, used to describe the growth of bacteria in batch suspensions, was not sound theoretically when applied to heterogeneous soils. I noticed immediately that many people were puzzled by my comment. After the talk, someone came to tell me that I had just thrown into the rubbish bin the only equation his teachers had ever managed to make him understand…

Moreover, part of the text gives the impression that soil physicist are superior to other soil scientists

> **Author's response**: Far from me the idea to set up soil physicists on a pedestal!!! In order to dispel any notion that I might want to do that, I have added a paragraph on lines 170 to 197 to that effect.

---

## Author Response (AR2)

**Itemized response to Editor's comments**

In this new round of revisions of the manuscript, I have accepted virtually all of the suggestions you made, as you can easily see from the file with Tracked changes. There are a few cases where I have decided not to follow the suggestions. They are discussed in the following.

On lines 59-63 of the previous version, you suggested changing the text to "To that end, given the sizeable task at hand, measurement methods were needed that at the same time were rapid, could be implemented directly in soil pits or in a rudimentary field laboratory without requiring sophisticated equipment, and *assumed that soil properties were static*, i.e., did not change appreciably over time."

> **Author's response**: I agree but I have tried to make the wording more accurate. Field pedologists did not assume soil properties to be static, but they zeroed in on a number of soil properties that, they thought, could be reasonable assumed to stay constant over extended periods of time. The text now reads (on lines 51-54 of the new version): "To that end, given the sizeable task at hand, measurement methods were needed that at the same time were rapid, could be implemented directly in soil pits or in a rudimentary field laboratory without requiring sophisticated equipment, and focused on soil properties that, one could reasonably assume, did not change appreciably over time."

On lines 110-113 of the latest version with your comments, you suggested that I write " Fortunately, the move towards "physics-informed machine learning (PIML), e.g. physics-informed neural networks" (Norouzi et al., 2025), which combines key soil water theory with machine learning approaches, signals a trend to move away from pure data-driven statistical approaches for generation of soil parameters."

> **Author's response**: I agree with the general idea behind this suggestion, but since I am not as enthused by these "physics-informed" proposals as you are, I have written that sentence slightly differently, as follows: "Also, some researchers have advocated for the use of "physics-informed" machine learning (PIML), and in particular of "physics-informed neural networks" (Wang et al., 2023; Norouzi et al., 2025), which perhaps signals a trend in the estimation of soil parameters to move away from pure data-driven statistical approaches toward attempting to combine some theory (in this case about soil water retention) with machine learning methods."

Similarly, on lines 145-149 of the latest version with your comments, you suggested that I include a sentence like: ".  In fact, the soil science community has got to be given credit for developing these hydro-thermal parameter maps (e.g. Montzka et al., 2017; Dai et al., 2019) as without them land surface modellers might still be using look-up tables of hydraulic and thermal parameters, with only a handful of soil types in them (Weihermueller et al., 2021). Also, these soil physical parameters feed into physical theories of water- and heat transfer in land surface and hydrological models."

> **Author's response**: The sentence I included in the text, on lines 104-107, reads: "One might argue in this respect that if hydro-thermal parameter maps (e.g. Montzka et al., 2017; Dai et al., 2019) had not been developed, land surface modellers might still be using look-up tables of hydraulic and thermal parameters, encompassing only a handful of soil types (Weihermueller et al., 2021)."

On line 190 of the latest version with your comments, you suggested that I add the sentence: "Fortunately, with physics-informed machine learning we can avoid these pitfalls (Wang, 2023).

> **Author's response**: The sentence I included in the text, on lines 142-144, reads: "Physics-informed machine learning (e.g., Wang et al., 2023; Norouzi, 2025), taking advantage of the fact that a theory exists to describe the retention of water by soils, may help alleviate some of these pitfalls. However, the problem persists for other dynamical aspects of soils for which no fundamental theory is as yet available."

On line 202 of the latest version with your comments, you questioned the expression "emphasis added".

> **Author's response**: This is a standard procedure when someone quoting a piece of text verbatim from another author, wants to add emphasis (in this case italics) to the text to underscore a part of it, when this emphasis was not present in the original text.